# Suppressing Alpha-Hemolysin as Potential Target to Screen of Flavonoids to Combat Bacterial Coinfection

**DOI:** 10.3390/molecules26247577

**Published:** 2021-12-14

**Authors:** Shangwen He, Qian Deng, Bingbing Liang, Feike Yu, Xiaohan Yu, Dawei Guo, Xiaoye Liu, Hong Dong

**Affiliations:** 1Beijing Traditional Chinese Veterinary Engineering Center and Beijing Key Laboratory of Traditional Chinese Veterinary Medicine, Beijing University of Agriculture, No.7 Beinong Road, Changping, Beijing 102206, China; shangwenhe@pku.edu.cn (S.H.); 202030311007@bua.edu.cn (Q.D.); 202030322107@bua.edu.cn (B.L.); 202030322117@bua.edu.cn (F.Y.); yu15512230962@163.com (X.Y.); godv2023@163.com (D.G.); 2Department of Mechanics and Engineering Science, College of Engineering, Peking University, No.5 Yiheyuan Road, Haidian, Beijing 100871, China

**Keywords:** flavonoids, bacterial coinfections, antitoxin, anti-hemolytic activity, inflammation

## Abstract

The rapid emergence of bacterial coinfection caused by cytosolic bacteria has become a huge threat to public health worldwide. Past efforts have been devoted to discover the broad-spectrum antibiotics, while the emergence of antibiotic resistance encourages the development of antibacterial agents. In essence, bacterial virulence is a factor in antibiotic tolerance. However, the discovery and development of new antibacterial drugs and special antitoxin drugs is much more difficult in the antibiotic resistance era. Herein, we hypothesize that antitoxin hemolytic activity can serve as a screening principle to select antibacterial drugs to combat coinfection from natural products. Being the most abundant natural drug of plant origins, flavonoids were selected to assess the ability of antibacterial coinfections in this paper. Firstly, we note that four flavonoids, namely, baicalin, catechin, kaempferol, and quercetin, have previously exhibited antibacterial abilities. Then, we found that baicalin, kaempferol, and quercetin have better inhibitions of hemolytic activity of Hla than catechin. In addition, kaempferol and quercetin, have therapeutic effectivity for the coinfections of *Staphylococcus aureus* and *Pseudomonas aeruginosa* in vitro and in vivo. Finally, our results indicated that kaempferol and quercetin therapied the bacterial coinfection by inhibiting *S. aureus* α-hemolysin (Hla) and reduced the host inflammatory response. These results suggest that antitoxins may play a promising role as a potential target for screening flavonoids to combat bacterial coinfection.

## 1. Introduction

The dissemination of bacterial infections accelerates the emergence of antibiotic tolerance [1,2,3]. However, current antibiotic therapy of bacterial infection leads to post-antibiotic expansion [4], and the resistant bacteria even facilitate the spread of resistance plasmids in the gut to cause multi-bacterial infections [5]. Additionally, the respiratory coinfections correlated with multiple bacterial pathogens are worse than the one with a single kind of bacterium in clinic [6,7,8]. Meanwhile, the increasing emergence of bacterial coinfections is paralyzing our public health systems worldwide [9,10]. Worse still, infections caused by mixed bacteria with unknown mechanisms are diminishing the discovery and development of new antibacterial drugs. Nevertheless, efficacious and novel antimicrobial agents remain the most effective strategy for the treatment of bacterial coinfections. Thus, there are urgent and unmet demands to identify novel or potential targets and to develop new antibacterial drugs with distinct modes of action to prevent such coinfections.

During coinfection, bacterial virulence factors frequently enhance the coinfections, such as where *Staphylococcus aureus* α-hemolysin (Hla) could cause the bacterial coinfections by potentiating the opportunistic bacterial infections [11]. In fact, *S. aureus* Hla belongs to the family of pore forming toxins (PFTs), which not only are involved in bacterial coinfections, but also associated with recurrent infections [11,12,13]. Therefore, we reasoned that the antitoxin ability of drugs might act as the drug screening principle for controlling bacterial coinfections. The plant natural flavonoids are ubiquitous in medicinal herbs and exhibit various properties on antitoxin and antibacterial activities [14,15,16]. Thus, flavonoids are utilized to screen drugs of antibacterial coinfections, which may offer alternatively the prospect of more effective therapies against bacterial coinfections through suppressing bacterial virulence factors. Four representative flavonoids including baicalin, catechin, kaempferol, and quercetin were selected to evaluate the therapeutic ability of bacterial coinfections. Among them, baicalin, kaempferol, and quercetin have an antitoxin ability. They target the hemolytic activity of *Staphylococcus aureus* Hla to combat the coinfection of *S. aureus* and *Pseudomonas aeruginosa*. Finally, kaempferol and quercetin indeed have therapeutic effectivity on bacterial coinfections in mouse lung. Since the aim of this paper is committed to finding potential drugs to combat bacterial coinfections, the data have shown that screening of flavonoids though the antihemolytic activity of Hla was feasible. We believe our work will shed light on the development of therapeutic strategies on bacterial coinfections in clinic.

## 2. Materials and Methods

### 2.1. Bacterial Strains and Culture Condition

The bacterial strains including *Staphylococcus aureus* ATCC29213 and *Pseudomonas aeruginosa* (a clinical strain isolated from cow dung) were used as the coinfection model strains in this study. Before infection, bacterial strains were cultured in Luria–Bertani (LB) broth (Beijing Aoxing Biotechnology Co., Ltd., Beijing, China) at 37 °C with agitation (220 rpm). For distinguish of the two strains, *Staphylococcus aureus* was cultured in mannitol sodium chloride agar (Aobox) and *Pseudomonas aeruginosa* was grown in pseudomonas Agar Base/CN Agar (Aobox).

### 2.2. Flavonoid Treatments and MIC Detection

Four kinds of flavonoids were employed in this paper. Flavonoids were dissolved in dimethyl sulfoxide (DMSO; Gibco; Shanghai, China) to a final concentration of 100 mg/mL as a drug storage fluid. More information of Flavonoids was detailed in Table 1.

### 2.3. Bacterial Growth Assays

Bacteria were incubated overnight in 3 mL of LB broth at 37 °C with shaking at 220 rpm. Then 100 μL bacteria (2 × 10^6^ CFU/mL) were added to a 96-well microtiter plate and treated with another 100 μL flavonoids at different concentrations. Finally, bacteria were incubated at 37 °C and the optical density was measured at 600 nm (OD_600nm_) for each two hours.

### 2.4. Hemolysis Assay

To evaluate the anti-hemolytic activity of flavonoids, the hemolysis assay was used for detection of the hemolytic activity of *S. aureus* Hla. Briefly, flavonoids were treated with *S. aureus* for 4 h and then the supernatants were collected and incubated into 5% fresh rabbit blood cells for about 30 min to detect hemolytic activity of the secreted *S. aureus* Hla. Then, PBS or 0.2% Triton X-100 were employed as negative or positive controls, respectively. The hemolysis rates of flavonoids were detected by using a plate reader (Tecan Infinite 200 pro) and calculating according to previous published work [1,17].

### 2.5. ELISA

The bacterial supernatants were collected to detect the levels of *S. aureus* Hla by ELISA (Shanghai Jijin Chemistry Technology Co., Ltd., Shanghai, China) according to the instructions.

### 2.6. Coinfection of S. aureus and P. aeruginosa In Vitro

The effect of flavonoids on coinfection of *S. aureus* and *P. aeruginosa* were evaluated by using the infected cellular model of pulmonary microvascular endothelial cells (PMVECs) according to the previous method [18,19]. Flavonoids including baicalin, catechin, kaempferol, and quercetin (at the concentration of 128 μg/mL) were used to treat with co-infected PMVECs, which 10^9^ cells per well were infected with *S. aureus* and *P. aeruginosa* at the final concentration ranges from 10^4^ to 10^8^ colony-forming units (CFUs) in DMEM with 1% FBS. Finally, the numbers of bacterial mixtures were quantified after 10 h and the hemolysis of mixtures were detected as per the previous method above.

### 2.7. Ethics Statement

Mice (eight-week mice) were purchased from the academy of military medical sciences, Beijing, China (Certificate Number: SCXK-PLA 2012-0004). All the experiments involved mice were gained an approval by the Institutional Animal Care and Use Committee at the Academy of Military Medical Sciences Institute (Beijing, China; approval no. SYXK2014-0002).

### 2.8. Lung Coinfection in Mice

Eight-week-old male Kunming mice were bred and maintained under specific-pathogen-free conditions. Mice were randomly divided into 9 groups of six each. Mice in the prevention group took 0.5% CMC-Na or flavonoids (100 mg/kg) by intragastric administration three days in advance. On the day of challenge, the mice were injected with a 1 × 10^9^ colony-forming units (CFUs) bacterial suspension or saline (0.05 mL/10 g) through the tail vein. After 2 h, 100 mg/kg of flavonoids or 0.5% CMC-Na were orally administered to mice in the infected treatment group. After 24 h infection, the mice were sacrificed by neck removal, and lung samples were collected. Left lungs were fixed in 4% formalin and submitted for histopathological sectioning and hematoxylin-eosin staining, and the samples were then visualized by light microscopy. Image J was used to analyze the number of inflammatory cells and count the numbers of the pulmonary alveolar area, while the PAA represented the proportion ratio of the pulmonary alveolar area to total area.

### 2.9. Western Blot

Primary pulmonary microvascular endothelial cells (PMVECs) were isolated from 1-day rats as per previous methods [1,18]. PMVECs were coinfected with *S. aureus* and *P. aeruginosa* as per the previous method above for 10 h. Then, the co-infected PMVECs were further treated with Hla or treated flavonoids for another 4 h. PMVECs were lysed by RIPA lysis-buffer (Beyotime, Shanghai, China) on ice for 15 min, and then the cell lysates were collected by centrifugation at 10,000× *g* for 20 min. The protein concentrations were quantified by a BCA kit (Beyotime, Shanghai, China). Then, proteins (30 μg) were separated by SDS-PAGE with 5–12% polyacrylamide gels and used to analyze the expressions of ASC, caspase-1, and NLRP3. The primary antibodies included rabbit anti-ACS, rabbit anti-caspase-1, and rabbit anti-NLRP3 (Abcam, Shanghai, China) and mouse anti β-actin (Proteintech, Beijing, China), secondary antibodies were goat anti-rabbit and goat anti-mouse antibodies (Abcam, Shanghai, China). The protein bands were quantified by ImageJ based on the gray values.

### 2.10. Statistical Analysis

Statistical analysis was performed using GraphPad Prism 8.0 (GraphPad Software). Data were expressed as the means ± standard deviations (SD). *p* values were calculated using unpaired or paired *t*-test between two groups. *p* > 0.05 were denoted “ns” and considered as no significant difference. All experiments were performed with at least 3 biological replicates. All animals were used for analysis unless the mice died.

## 3. Results

### 3.1. Screening of Flavonoids by the Antihemolytic Activity

Flavonoid compounds have a basic skeleton with the 2-phenyl-chromone with C6-C3–C6 system (Graphic picture). The antibacterial effects of flavonoids have been previously reported [20,21]. Four typical flavonoids including baicalin, catechin, kaempferol, and quercetin (Table 1), which were abundant in plants were selected to assess the direct action to *Staphylococcus aureus*. Among them, baicalin, which has been shown to have the better effect on bacterial toxins, was used as a reference to screen flavonoids [22]. Results showed that all the four flavonoids have antibacterial activities (Figure 1A), especially in the 8 h treatment (Figure 1B). Subsequently, the antihemolytic activity of flavonoids against *S. aureus* Hla were detected (Figure 2A); results showed that catechin did not have any antihemolytic activity under the concentrations of 8–128 μg/mL, whereas baicalin, quercetin, and kaempferol could inhibit the hemolysis of Hla (Figure 2B,C). In addition, these flavonoids exhibited different levels of antihemolytic activity, namely, the effective concentration of kaempferol began at 8 μg/mL treatment, quercetin 32 μg/mL, and baicalin at 128 μg/mL (Figure 2C). Further, to verify whether the flavonoids inhibited the hemolysis of Hla, depending on directly targeting Hla secretion from *S. aureus*, we quantified the amount of Hla by ELISA. As shown in Figure 2D, the similar tends of Hla levels under the treatment of flavonoids were detected compared to the antihemolytic activity. Altogether, our results suggested that the antihemolytic activity of flavonoids dominated the inhibition of Hla levels. Based on this screening principle, we selected two flavonoids included kaempferol and quercetin as having the best efficacy.

### 3.2. Flavonoids Combat the Coinfection of Staphylococcus aureus and Pseudomonas aeruginosa by Targeting Hla

To assess whether the flavonoids impacted on bacterial expansion in vitro. Firstly, bacterial suspensions of *S. aureus* and *Pseudomonas aeruginosa* were infected with pulmonary microvascular endothelial cells (PMVECs) in different proportions from 10^4^ to 10^8^ CFU/mL, results suggesting that kaempferol and quercetin have better inhibitions of bacterial coinfections (Figure 3A). Then, the bacterial supernatants were collected to determine the antihemolytic activity (Figure 3B). Date showed that baicalin, kaempferol, and quercetin significantly reduced the levels of Hla, revealing that flavonoids inhibited the growth of mixed bacteria and might be regulated by suppressing Hla (Figure 3C). Overall, these findings confirming that inhibition of Hla can decrease bacterial coinfections.

To further investigate the therapeutic efficacy of flavonoids against bacterial coinfections in vivo, a lung infective model was employed to detect (Figure 4A). Briefly, mice were infected with 10^9^ CFUs of *S. aureus* and *P. aeruginosa*. Then, mice were pretreated with flavonoids for 3 days and then co-infected with bacteria, or the co-infected mice were treated with flavonoids after 2 h coinfection. Finally, pathological and antibacterial analysis was carried out at 5 days for pretreated groups and 24 h for treated groups. The quantification of bacteria in lung tissues showed that kaempferol has a more curative effect than quercetin either in pretreatment or treatment of flavonoids (Table 2, Figure 4B,C). This is consistent with the results of antihemolytic activity between the two flavonoids, which showed that kaempferol was also superior to quercetin (Figure 2C). It is worth noting that the bacterial loading of *S. aureus* in mouse lung from the other groups showed no significant changes compared to the untreated group, only the numbers of *P. aeruginosa* from coinfected mice were decreased under kaempferol pretreatment or treatment (Figure 4B,C), probably due to down-regulation effect of kaempferol on Hla levels of *S. aureus* to reduce dissemination of *P. aeruginosa*. Altogether, these data suggest that the better inhibition of bacterial virulence factor, the better anti-coinfection of flavonoids.

### 3.3. Flavonoids Therapy the Bacterial Coinfection Require to Reduce the Host Inflammatory Response to Assist

To investigate the treatment differences between kaempferol and quercetin on combating bacterial coinfection, the pathological detection of co-infected lung tissues was analyzed by H&E staining. Images showed that the increased inflammatory cells (black arrowheads) accompanied with congestion points (blue arrowheads) in the lung tissue were only found from co-infected mice compared to *S. aureus* or *P. aeruginosa* alone infections (Figure 5A(upper),B). In addition, the *P. aeruginosa* infection and bacterial coinfection decrease the pulmonary alveolar area (PAA), suggesting that damage of lung tissue occurred (Figure 5C). Additionally, pretreatment or treatment of flavonoids with co-infected mice showed that kaempferol and quercetin had the pretreated function for bacterial coinfection, while only kaempferol could treat the co-infected mice (Figure 5A,D,E). These findings indicate that flavonoids decrease the tissue damage, and inflammation might have assisted the therapy of bacterial coinfection. 

Previously, we had confirmed that flavonoids therapied the bacterial coinfection though suppressing Hla (Figure 3). Thus, we further investigate the impact of flavonoids on inflammatory pathway of the Hla-treated PMVECs. Firstly, we found that Hla indeed enhanced the inflammation of co-infected PMVECs by upregulations of the inflammatory proteins of ASC, caspase-1, and NLRP3 (Figure 6A,B), which might explain why the coinfected mice had larger numbers of inflammatory cells (Figure 5A,B). Next, kaempferol and quercetin significantly reduced the inflammation of the co-infected PMVECs in a dose-dependent manner (Figure 6A,B), suggesting that inhibition of the inflammation was required in the antiinfection strategy of flavonoids. Lastly, we further detected the inflammatory factors of IL-1β and IL-18; results also revealed that kaempferol and quercetin could decrease the inflammation (Figure 6C). Taken together, our results illustrate that to what extent the flavonoids therapied the bacterial coinfection, depended on the decreased inflammation.

## 4. Discussion

Flavonoid compounds contain oxygen heterocyclic ring and its basic skeleton is 2-phenylchromone. They are widely found in various Chinese herbal medicines and have good antibacterial and anti-inflammatory activities. Most flavonoids, such as luteolin, kaempferol, licochalcone A, apigenin, quercetin, and catechins, have been reported as having an antibacterial effect [23,24]. This role is currently one of the important directions for research and development of new antibacterial drugs and new food additives [25]. Baicalin is a traditional Chinese medicine component of flavonoids extracted from *Scutellaria baicalensis Georgi.* In clinical studies, it has been found to have multiple biological activities, namely, anti-tumor [26], anti-oxidation [27], anti-virus [28], antibacterial, and anti-inflammatory [29]. Among them, in addition to direct antibacterial aspects, studies have shown that baicalin can produce synergistic antibacterial activity against *Staphylococcus aureus* by inhibiting NorA efflux protein with oxacillin, tetracycline, and ciprofloxacin [15,30]. Reports show that antibacterial activity of catechins isolated from cashew nut shells and found that catechins have the potential to resist methicillin-resistant *Staphylococcus aureus* (MRSA), which can increase the production of reactive oxygen species and reduce the oxidative stress of antioxidant enzymes [31]. Kaempferol is found in many natural plants such as *Bupleurum*, *Eucommia*, *Hippophae rhamnoides*, *Ginkgo biloba*, *Panax notoginseng*, mulberry branches, and acacia flowers. Kaempferol 3-rutanoside isolated from *acacia flowers* can inhibit the adhesion and invasion of *Streptococcus mutans* to the host by inhibiting the activity of sortase A and exert antibacterial effects [32]. The antibacterial activity of different flavonoids was related to lipophilicity. Reports showed that kaempferol can cause damage to bacterial cells by interacting with the polar head group of the model cell membrane [33,34]. Quercetin is mainly distributed in loquat, rhodiola, raspberry, chrysanthemum, sweet dices, and forsythia. In the study of nine flavonoids from the leaves of *Scutellaria baicalensis Georgi*, it was found that quercetin-3-glucoside can inhibit *Staphylococcus aureus* and reduce biofilm formation [35]. In addition, it was found in another study that the combination of quercetin and amoxicillin has synergistic antibacterial activity against amoxicillin-resistant *Staphylococcus epidermidis* [36]. When observing the inhibitory effect of quercetin on oral microorganisms against Gram-negative bacteria, it was found that quercetin had strong antibacterial activity against *Porphyromonas gingivalis*, and its minimum inhibitory concentration is 0.0125 μg/mL [37]. In this article, we measured the minimum inhibitory concentration of four flavonoids in traditional Chinese medicine and found that the MICs of baicalin and catechin against *Staphylococcus aureus* were 128 μg/mL and 1024 μg/mL, respectively. The MIC of kaempferol and quercetin was greater than 2048 μg/mL. However, in the bacterial growth heat map, we found that kaempferol and quercetin could better inhibit the growth of *S. aureus* in a concentration-dependent manner in the early stage of co-culture and have a certain antibacterial effect.

Current studies have found that Hla can enhance the proliferation and spread of Gram-negative bacteria by preventing acidification of phagosomes of bacterial-containing macrophages, which is a key factor in causing mixed bacterial infections [11]. Therefore, on the basis of the previous research, we explored whether the flavonoids with better inhibitory effect on Hla could inhibit the coinfection of *S. aureus* and *P. aeruginosa*. First, the Punnett square was used to observe the direct inhibitory effect of the flavonoids on the mixed culture in vitro. The results showed that baicalin, catechin, kaempferol, and quercetin had better inhibitory effects on mixed culture. Of note, we found that kaempferol showed stronger antibacterial activity when the ratio of *S. aureus* in the mixed culture decreased, suggesting that kaempferol has a strong antibacterial activity against *P. aeruginosa*. Moreover, when counting the number of lung colony in the coinfection mice model, it was found that both kaempferol treatment and prevention group could reduce the number of *P. aeruginosa* in lung tissue, indicating that kaempferol could effectively inhibit the growth of *P. aeruginosa* in vivo and in vitro. Mourouge et al. found that Kaempferol-3-o-(2′,6′-di-o-trans-p-coumaryl)-β-d-glucopyranoside (KF) isolated from *Melastomataceae*, which can inhibit *P. aeruginosa* by inhibiting the expression of virulence factors, affects its nutrient absorption, movement, and growth mechanism when they studied the use of plants to treat acute *P. aeruginosa* infections, indicating that kaempferol has great potential in the inhibition of *P. aeruginosa* [38].

A mice model of bacterial coinfection was established by tail vein injection. The weight changes in mice before and after infection were counted, and lung tissue colony counts, lung pathological sections, and serum inflammatory factor levels were tested to observe the success of the coinfection model and evaluate the protective effects of kaempferol and quercetin on mixed infected mice. The results showed that the coinfection group caused higher body weight changes than the single infection group, it also promoted the growth of *S. aureus* and *P. aeruginosa* in the lungs, causing more inflammation than a single infection. This is consistent with the results of a previous Cohen study that *S. aureus* can enhance the proliferation and lethality of Gram-negative bacteria [11]. This showed that the inflammation model of coinfection was successfully established. In the use of flavonoids for prevention and treatment, we found that kaempferol and quercetin, which have good inhibitory effect on Hla, can reduce the weight change in mice before and after infection, it can reduce the accumulation of *P. aeruginosa* in lung tissue and lung injury, and inhibit the expression of inflammatory factors, which has a protective effect on coinfection mice. These results indicate that flavonoids play a promising role in combating bacterial coinfection and they also provide a reference for the development and utilization of plant extracts rich in flavonoids in the future.

## Figures and Tables

**Figure 1 molecules-26-07577-f001:**
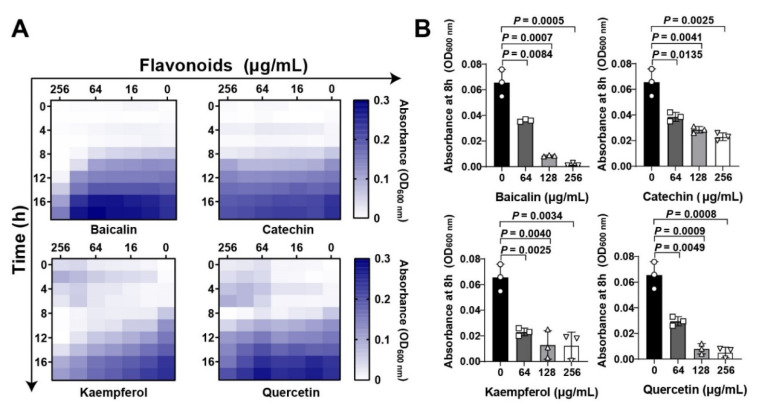
Flavonoids inhibit *Staphylococcus aureus* growth at the initial period: (**A**) Flavonoids inhibited *S. aureus* in both time- and dose dependent manners. Chessboard method was employed to detect the antibacterial effect of flavonoids. Different concentrations of flavonoids including baicalin, catechin, kaempferol, and quercetin (0 to 2048 μg/mL) were treated with *S. aureus* at different time points (0, 4, 8, 12, 16, and 18 h). Bacterial growth was quantified based on the absorbance by a plate reader at wavelength of 600 nm. (**B**) Quantification of *S. aureus* absorbance at 8 h under flavonoid treatment at the final concentrations of 0, 64, 128, and 256 μg/mL. Data were shown as Mean ± SED (n = 6).

**Figure 2 molecules-26-07577-f002:**
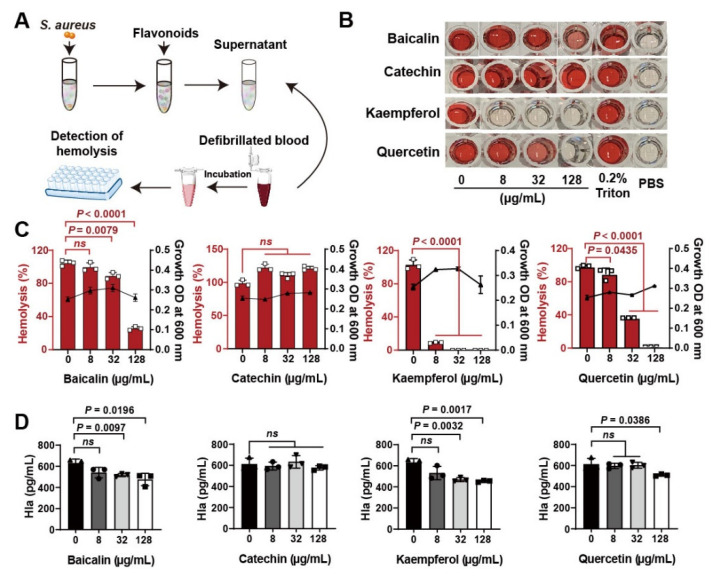
Screening of flavonoids on the hemolytic activity of *S. aureus* Hla: (**A**) Detection of hemolytic activity workflow. (**B**) Images of flavonoids treated with *S. aureus* Hla in blood cell suspension. The supernatants of *S. aureus* treated with different concentrations of flavonoids (0–128 μg/mL) incubated with the rabbit blood cell suspension for 20 min. (**C**) Quantification of hemolytic activation of flavonoids. (**D**) Detection of Hla levels by ELISA. Experiments were performed with at least 3 independent repeats. *ns* indicates *p* > 0.05 showed no significant differences.

**Figure 3 molecules-26-07577-f003:**
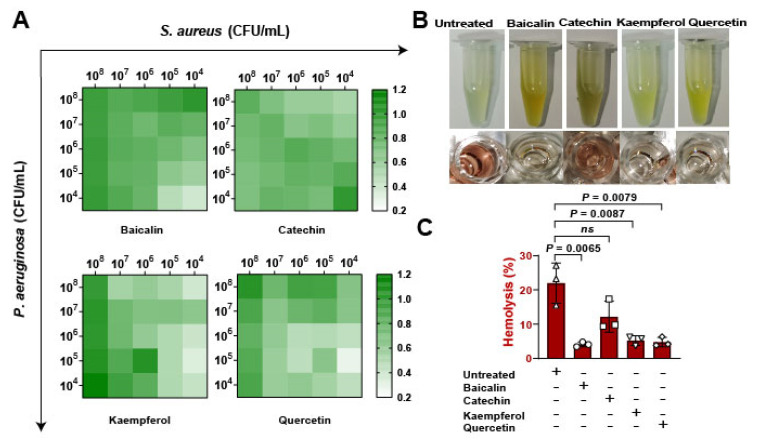
Detection of flavonoids on the growth of *S. aureus* and *P. aeruginosa* in vitro: (**A**) The antibacterial function of flavonoids against the coinfection of *S. aureus* and *P. aeruginosa.* Pulmonary microvascular endothelial cells (PMVECs) were infected with the mixture of *S. aureus* and *P. aeruginosa* at different CFUs (10^4^ to 10^8^ CFU/mL) and treated with flavonoids including baicalin, catechin, kaempferol, and quercetin (at the concentration of 128 μg/mL) for 10 h. (**B**) Images of the hemolytic activity of bacterial mixtures of *S. aureus* (1 × 10^8^ CFUs) and *P. aeruginosa* (1 × 10^8^ CFUs) treated with flavonoids for 10 h. (**C**) The hemolysis detections of the mixtures of *S. aureus* and *P. aeruginosa* after the flavonoid treatment.

**Figure 4 molecules-26-07577-f004:**
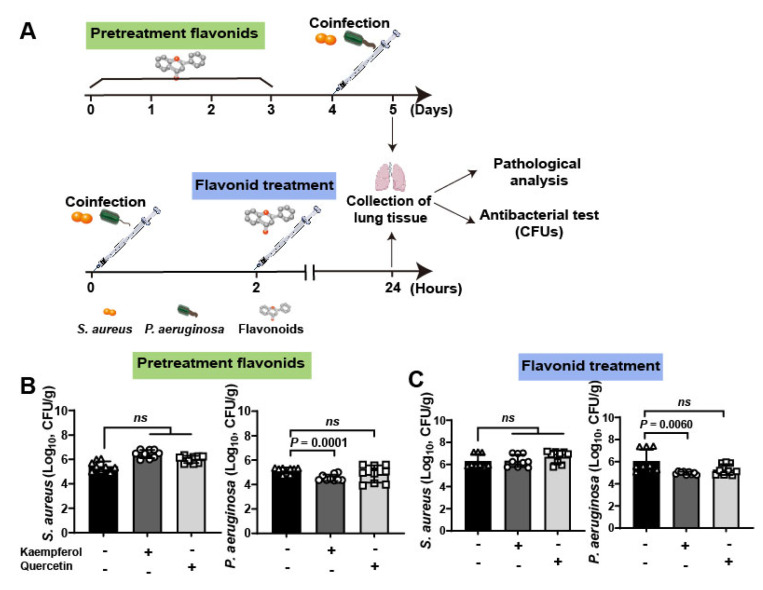
Clinical validation of flavonoids on the mouse lung coinfection of *S. aureus* and *P. aeruginosa*: (**A**) Scheme of the pre-treatment and treatment of flavonoids on mouse lung coinfection. Briefly, mice were coinfected with *S. aureus* (1 × 10^9^ CFUs) and *P. aeruginosa* (1 × 10^9^ CFUs) and flavonoids were pre-treated or treated with infected mice for further determine the therapeutic effects. (**B**,**C**) Kaempferol had the better antibacterial effects on the lung coinfection of *S. aureus* and *P. aeruginosa* compared to quercetin on both pre-treatment (**B**) and treatment (**C**).

**Figure 5 molecules-26-07577-f005:**
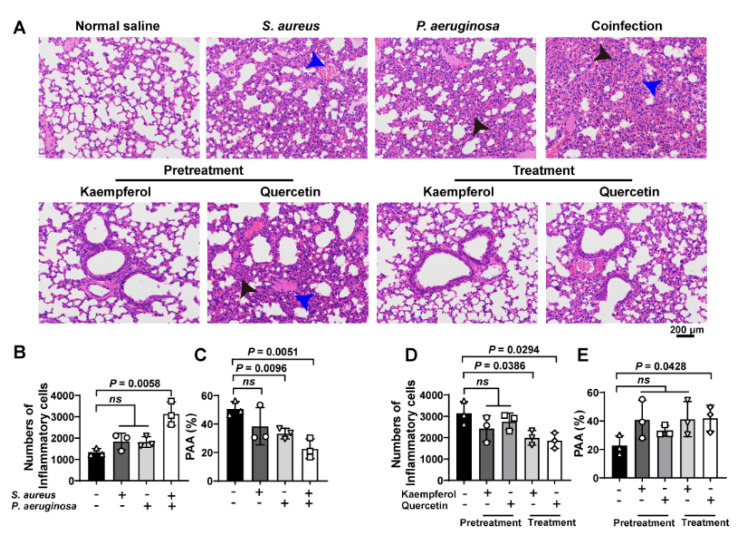
Flavonoids have a better therapeutic effect on lung coinfections of *S. aureus* and *P. aeruginosa* in vivo: (**A**) H&E staining of the lung tissue. Mice were infected with *S. aureus*, *P. aeruginosa* or the mixture of *S. aureus* and *P. aeruginosa.* Then, infected mice were treated with flavonoids. The black arrows show inflammatory cell infiltration. The blue arrows indicate the structure of the alveolar cavity. Scar bar = 200 μm. (**B**–**E**) Quantitative analysis of inflammatory cells number and pulmonary alveolar area (PAA) in mouse lung tissues. (**B**,**C**) Quantification of the number of inflammatory cells and the PAA in the lung tissues of infected mice. (**D**,**E**) Quantitative analysis of pulmonary inflammatory cells and PAA after prevention or treatment with kaempferol or quercetin. Three independent experiments were performed to obtain stable results. *ns* indicated no significant difference when compared with the non-treated controls.

**Figure 6 molecules-26-07577-f006:**
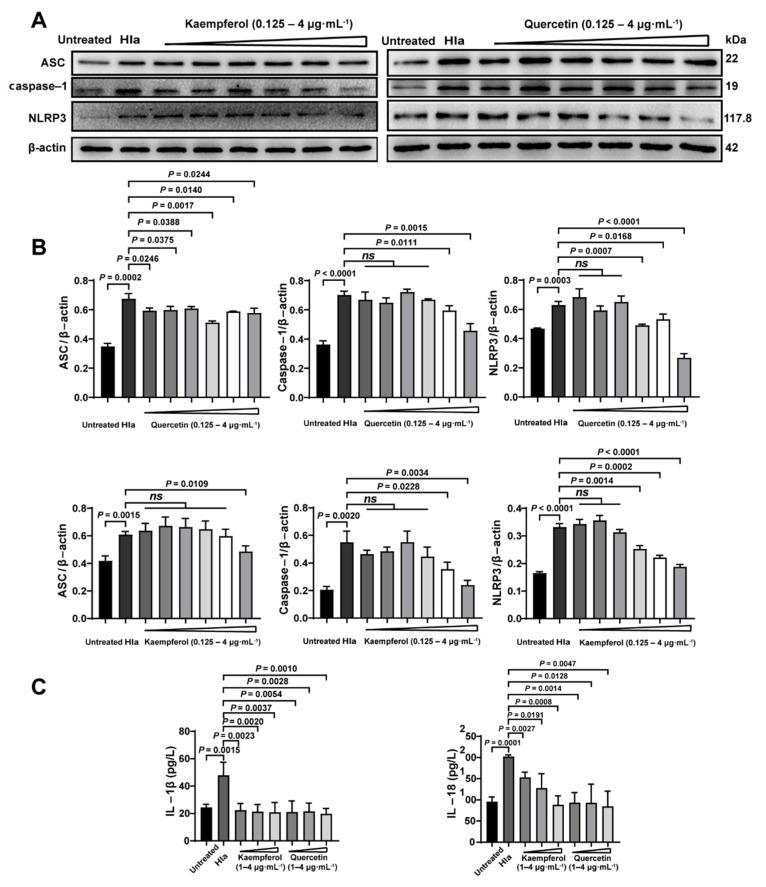
Flavonoids decreased the inflammatory levels caused by Hla in vitro detection: Pulmonary microvascular endothelial cells (PMVECs) were treated Hla with 4 h (**A**) ASC, Caspase-1 and NLRP3 expressions of Kaempferol and quercetin treatment were detected by Western Blot. (**B**) The corresponding proteins analyzed from A were normalized to the of levels of β-actin. (**C**) The IL-1β and IL-18 levels were determined by ELISA assay.

**Table 1 molecules-26-07577-t001:** Flavonoids used in this study.

Flavonoids	Molecular Formula	Chinese Herbs	Classic Prescription	Source
Baicalin	C_21_H_18_O_11_	*Scutellaria baicalensis Georgi*	Qingying Decoction	Beijing Solarbio Science & Technology Co., Ltd. (Beijing, China)
Catechin	C_15_H_14_O_6_	*Paeoniae Radix Rubra*	Xijiao Dihuang decoction	
Kaempferol	C_15_H_10_O_6_	*Bupleurum chinense*	Bupleurum chinense	Shanghai yuanye Bio-Technology Co., Ltd. (Shanghai, China)
Quercetin	C_15_H_10_O_7_	*Forsythiae Fructus*	Qingying Decoction	

**Table 2 molecules-26-07577-t002:** Flavonoid treatment had a significant effect on body weight but no effect on organ index in co-infected mice.

Group	Weight (g)	Index (%)				
		Heart	Liver	Spleen	Lung	Kidney
Saline solution	1.42 ± 0.60	0.78 ± 0.21	5.75 ± 0.30	0.44 ± 0.03	0.66 ± 0.12	1.48 ± 0.11
*S. aureus*	4.64 ± 1.94 ^##^	0.69 ± 0.12	5.74 ± 0.44	0.58 ± 0.09	0.78 ± 0.06	1.54 ± 0.06
*P. aeruginosa*	3.67 ± 2.27 ^#^	0.82 ± 0.10	6.44 ± 0.57	0.64 ± 0.06 ^##^	0.75 ± 0.07	1.70 ± 0.15
Coinfection	6.02 ± 0.85 ^###^	0.69 ± 0.11	6.10 ± 0.49	0.47 ± 0.06	0.75 ± 0.02	1.78 ± 0.19
Coinfection with Kaempferol treatment	3.77 ± 1.96 *	0.63 ± 0.28	6.64 ± 1.02	0.47 ± 0.07	0.76 ± 0.03	1.63 ± 0.17
Coinfection with Quercetin treatment	3.81 ± 1.18 **	0.67 ± 0.10	6.59 ± 0.45	0.47 ± 0.04	0.79 ± 0.07	1.73 ± 0.10

Note: ^#^
*p* < 0.05, ^##^
*p* < 0.01, ^###^
*p* < 0.001 compared with the uninfected group. * *p* < 0.05, ** *p* < 0.01 compared with the co-infection group (n = 6).

## Data Availability

Not applicable.

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
