# Peer review of "Suppressing Alpha-Hemolysin as Potential Target to Screen of Flavonoids to Combat Bacterial Coinfection"

_molecules, 2021, doi:10.3390/molecules26247577_

Round 1

Reviewer 1 Report

Suppressing alpha-haemolysin as potential target to screen of flavonoids to combat bacterial coinfection

The authors hypothesized that antitoxin hemolytic activity was served as the screening principle to select antibacterial drugs to combat coinfection from natural products

I suggest accepting the work but is necessary Extensive editing of the English language and style

examples:

in table 1 the species name must be in italics

in line 269… “Scutellaria baicalensis” Georgi. name must be in italics

in line 279 “Hippophae rhamnoides, Ginkgo biloba, Panax notoginseng species name must be in italics

in line 287 ..”Scutellaria baicalensis” Georgi, it was found that quercetin-3-glucoside can inhibit “Staphylococcus aureus” name must be in italics!!

in line 292 “Porphyromonas gingivalis” name must be in italics!!

in line 151. Revise …. The antibacterial activates, change by antibacterial activities

Author Response

Thank you for your detailed comments, which make our writing more standardized. We have revised our manuscript and prepared a point-to-point response to the comments on above responses. All changes are highlighted by yellow in the revised manuscript.

Q1: in table 1 the species name must be in italics.

Response: Thank you for your suggestions. According to your comments, we have revised the species names by using italics in Table 1 and throughout the manuscript.

Q2: in line 269… “Scutellaria baicalensis” Georgi. name must be in italics.

Response: Thanks for your comments. We have already revised it.

Q3: in line 279 “Hippophae rhamnoides, Gin kgo biloba, Panax notoginseng species name must be in italics.

Response: Thank you for your suggestions. We have changed it on Page 10, line, 283-284.

Q4: in line 287 ..”Scutellaria baicalensis” Georgi, it was found that quercetin-3-glucoside can inhibit “Staphylococcus aureus” name must be in italics!!

Response: Thanks for your comments. We have already revised it on Page 10, line, 290-292.

Q5: in line 292 “Porphyromonas gingivalis” name must be in italics!!

Response: Thanks for your suggestions. We have changed it on Page 10, line, 296.

Q6: in line 151. Revise …. The antibacterial activates, change by antibacterial activities.

Response: Thanks for your comments. We have changed the " antibacterial activates " to " antibacterial activities " on Page 4, line 154.

Reviewer 2 Report

Very interesting work. There are only a couple of objections (comments).

1. How were these 4 flavonoids selected? Why only 4?
2. Haven't you considered using a plant extract that is rich in flavonoids instead of these standards?

Author Response

Reviewer 2:

Thank you for recognizing the significance of our work. We are grateful for the insightful comments. Having now gone through these points in depth, we have revised our manuscript and highlighted the changes by yellow in the revision. A point-by-point response to the comments is attached.

Q1: How were these 4 flavonoids selected? Why only 4?

Response: Thank you for your insightful comments. The plant natural flavonoids are ubiquitous in medicinal herbs and exhibit various properties on antitoxin and antibacterial activities (Acta Pharmaceutica Sinica B 2016, 6, 148-157 Adv Sci 2021, 8, e2100749; Phytotherapy Research 2014, 28, 1071-1076). Due to the flavonoid compounds have a basic skeleton with the 2-phenyl-chromone with C6-C3-C6 system, we selected four representative ones based on the structures (Figure R1; See also Graphic picture). As showed in the right panel of Figure R1, these four representative flavonoids including baicalin, catechin, kaempferol and quercetin were selected to evaluate the therapeutic ability of bacterial coinfections. These results indicated that flavonoids play a promising role in combating bacterial coinfection and these also provided the reference for the development and utilization of plant extracts rich in flavonoids in the future. These illustrations on the selected principle of the four flavonoids have also added in the revised manuscript on Pages 2 and 11.

Figure R1. The structure of four flavonoids

Q2: Haven't you considered using a plant extract that is rich in flavonoids instead of these standards?

Response: Thank you for the constructive comment. Indeed, we carry out the present work aims to reveal that whether the antitoxin ability of drugs might act as the drug screening principle for controlling bacterial coinfections and we selected four typical flavonoids as model drugs to test. For this purpose, these flavonoid standards are more suitable for the present work due to the determined component and structure. Therefore, the development of the plant extracts that rich in flavonoids are our future work to discover and screening protential drugs for therapy of bacterial coinfections.

Reviewer 3 Report

The manuscript is very interesting and providing the original images from blots and gels is highly appreciated.

The authors need to use italics for all species names used in the paper. Also, the authors have to move the references before punctuation at the end of the sentence, respectively to use space before brackets of references in the text. 

Author Response

Reviewer 3:

Thanks for your comments. We have revised our manuscript (highlighted in yellow in the main text) and prepared a point-to-point response to the comments on above responses. I hope the revised manuscript will comply with the criteria of the journal.

Q1: The authors need to use italics for all species names used in the paper.

Response: Thank you for your detailed suggestions. We have revised the species names by using italics throughout the manuscript.

Q1: the authors have to move the references before punctuation at the end of the sentence, respectively to use space before brackets of references in the text.

Response: Thanks for your comment. We have already revised the format of all references in the main text in the revised manuscript.